# No-Reference Hyperspectral Image Quality Assessment via Ranking Feature Learning

Yuyan Li [1,†] , Yubo Dong [2,3,†], Haoyong Li [4,*], Danhua Liu [2,5], Fang Xue [6] and Dahua Gao [2]

1  Guangzhou Institute of Technology, Xidian University, Guangzhou 510555, China; liyuyan@stu.xidian.edu.cn
2  School of Artificial Intelligence, Xidian University, Xi'an 710068, China; ybdong@stu.xidian.edu.cn (Y.D.); dhliu@xidian.edu.cn (D.L.); dhgao@xidian.edu.cn (D.G.)
3  School of Electrical and Electronic Engineering, Nanyang Technological University, Singapore 639798, Singapore
4  Academy of Advanced Interdisciplinary Research, Xidian University, Xi'an 710071, China
5  Pazhou Lab, Huangpu 510555, China
6  Beijing Institute of Space Mechanics and Electricity, Beijing 100094, China; xuefang_007@163.com
*  Correspondence: lihaoyong@xidian.edu.cn
†  These authors contributed equally to this work.

**Abstract:** In hyperspectral image (HSI) reconstruction tasks, due to the lack of ground truth in real imaging processes, models are usually trained and validated on simulation datasets and then tested on real measurements captured by real HSI imaging systems. However, due to the gap between the simulation imaging process and the real imaging process, the best model validated on the simulation dataset may fail on real measurements. To obtain the best model for the real-world task, it is crucial to design a suitable no-reference HSI quality assessment metric to reflect the reconstruction performance of different models. In this paper, we propose a novel no-reference HSI quality assessment metric via ranking feature learning (R-NHSIQA), which calculates the Wasserstein distance between the distribution of the deep features of the reconstructed HSIs and the benchmark distribution. Additionally, by introducing the spectral self-attention mechanism, we propose a Spectral Transformer (S-Transformer) to extract the spatial-spectral representative deep features of HSIs. Furthermore, to extract quality-sensitive deep features, we use quality ranking as a pre-training task to enhance the representation capability of the S-Transformer. Finally, we introduce the Wasserstein distance to measure the distance between the distribution of the deep features and the benchmark distribution, improving the assessment capacity of our method, even with non-overlapping distributions. The experimental results demonstrate that the proposed metric yields consistent results with multiple full-reference image quality assessment (FR-IQA) metrics, validating the idea that the proposed metric can serve as a substitute for FR-IQA metrics in real-world tasks.

**Keywords:** hyperspectral; image assessment; no reference; deep feature

## 1. Introduction

Hyperspectral images (HSIs) contain detailed scene representation information and have a wide range of applications in various fields, including remote sensing [1–3] and object detection [4–6]. To acquire high-quality HSIs, many HSI restoration methods have been proposed, such as HSI reconstruction [7–12], HSI denoising [13–15], and HSI super-resolution [16–18]. Considering HSI reconstruction, coded aperture snapshot spectral imaging (CASSI) can achieve fast imaging by multiplexing a 3D HSI into a 2D measurement [19–23], and the 2D measurement can be reconstructed to HSIs by reconstruction algorithms. As shown in Figure 1, due to the lack of ground truth in real imaging processes, these reconstruction methods are usually trained and validated on simulation datasets and then tested on real measurements captured by real HSI imaging systems. However, due to the gap between the simulation imaging process and the real imaging process, the best

model validated on the simulation dataset may fail on real measurements. Therefore, to obtain the best model in a real-world task, it is crucial to design a suitable no-reference HSI quality assessment metric to reflect the reconstruction performance of different models.

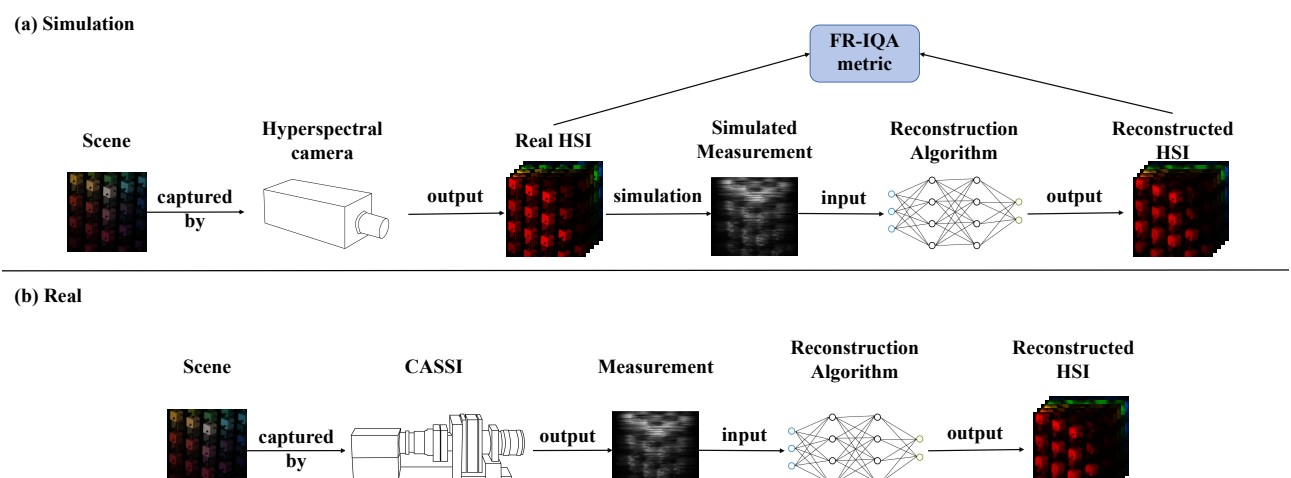

**Figure 1.** The flowchart of HSI reconstruction task. (**a**) is simulation task. (**b**) is real-world task.

Currently, there are few no-reference image quality assessment methods for HSIs. Previous work proposed a no-reference hyperspectral image quality assessment method based on quality-sensitive statistical features (QSFL) [24] that uses statistical methods to extract manual features and utilizes the Bhattacharyya distance between the distribution of manual features of the reconstructed image and the benchmark distribution as the evaluation score. However, the manual feature extraction process was laborious, and the extracted features are less representative. Deep features have demonstrated stronger representational and generalization capabilities compared to manual features [25–31]. However, these methods were developed for RGB images and neglect the spectral information of HSIs, which is crucial for evaluating spectral fidelity. To address these issues, we propose a Spectral Transformer (S-Transformer) based on Spectral-wise Multi-head Self-Attention (S-MSA) [32] to extract deep features, which can effectively model the interspectra similarity and correlations of HSIs.

To extract quality-sensitive features, a reasonable pre-training task is necessary. Since humans cannot visually evaluate the quality of the spectrum, there is a lack of HSI datasets with Mean Opinion Score (MOS) labels for HSI quality assessment. To address this issue, we employ the unsupervised ranking feature learning task [28] for pre-training. The ranking feature learning task ranks pairs of images according to their quality, enabling the deep network to better capture the quality-related differences between images. In this approach, the S-Transformer can effectively extract quality-sensitive deep features. Additionally, the ranking feature learning task also increases the scale of the dataset, further improving the performance and generalization capability of the model.

QSFL [24] used the modified Bhattacharyya distance to evaluate the difference between the distribution of the reconstructed HSIs and the benchmark distribution. However, when the distributions are completely non-overlapping, the Bhattacharyya distance may not provide a meaningful comparison. To address this issue, we introduce Wasserstein distance [33] to measure the distance between the distribution of the deep features of the reconstructed HSIs and the benchmark distribution. By introducing the Wasserstein distance, the discrepancy between the distributions of the deep features can be more accurately reflected, thereby enhancing the assessment ability of the proposed method.

In summary, we propose a no-reference HSI quality assessment metric. The proposed method pre-trains the S-Transformer to extract quality sensitive deep features via ranking feature learning. Then, the Wasserstein distance between the distribution of the deep fea-

tures of the reconstructed HSIs and the benchmark distribution is calculated as the quality assessment score.

Our contributions can be summarized as follows:

- We propose a novel no-reference quality assessment metric for HSIs. Currently, there are few no-reference quality assessment methods for HSIs. To the best knowledge of the authors, the proposed metric is the first method that uses deep features for HSI quality assessment.
- A S-Transformer is proposed. The proposed S-Transformer is designed to extract deep features based on the characteristics of HSIs, which could capture the interspectral similarity of HSIs through spectral self-attention.
- We choose ranking feature learning as the pre-training task of the S-Transformer. The ranking feature learning task ranks pairs of images according to their quality, enabling the S-Transformer to better capture the quality-related differences between images.
- The Wasserstein distance is introduced to measure the distance between the distributions of the deep features. By introducing the Wasserstein distance, the discrepancy between the distributions of the deep features can be more accurately reflected, thereby enhancing the assessment ability of the proposed method.

## 2. Materials and Methods

### 2.1. Related Work

2.1.1. Full-Reference Hyperspectral Image Quality Assessment

Full-reference image quality assessment (FR-IQA) metrics evaluate the performance of reconstruction algorithms by calculating the similarity between ground truths and reconstructed HSIs. Spatial fidelity and spectral fidelity are both important for evaluating the quality of HSIs.

The Peak Signal-to-Noise Ratio (PSNR) and Structural Similarity Index (SSIM) [34] are common spatial fidelity assessment metrics for HSIs. SSIM is originally proposed for greyscale images, and HSI reconstruction works usually calculate SSIM band-by-band and then average them [7–12] to obtain the SSIM score of HSIs. In [35], Multivariate SSIM (MvSSIM) is proposed to better generalize SSIM to HSIs by replacing the univariate sampling statistics in SSIM with their multivariate versions. Samiran et al. proposed an ensemble FR-IQA metric to quantify the quality of HSIs based on spatial features in [36].

Spectral Angle Mapping (SAM) [37] is a common spectral fidelity FR-IQA metric for HSIs. It reflects the similarity of spectral curves between ground truths and reconstructed HSIs. Recently, $Q2^n$ [38] was proposed to index the quality of multispectral and hyperspectral images based on the computation of the hypercomplex correlation coefficient between the reference and tested images, which jointly measures spectral and spatial distortions.

2.1.2. No-Reference Image Quality Assessment

There are few no-reference HSI quality assessment methods. QSFL [24] is the only available no-reference HSI quality assessment method. QSFL computes MSCN coefficients in the spatial and spectral dimensions separately and extracts GMSD coefficients using Log-Gabor filters. These coefficients are fitted to a Gaussian distribution model and composed the statistical features. The quality score is obtained by calculating the Bhattacharyya distance between the distribution of statistical features and the benchmark distribution. QSFL [24] demonstrated its effectiveness by comparing the scores of the original images with the reconstructed images from state-of-the-art HSI super-resolution methods.

There are some no-reference image quality assessment (NR-IQA) methods for remote sensing images, pan-sharpening, and RGB images. These NR-IQA methods are either based on statistical features or deep features. Some researchers present NR-IQA methods based on statistical features for pan-sharpening images [39–41]. Deep features are used in some NR-IQA methods for pan-sharpening images [42,43]. An NR-IQA method based on deep features for remote sensing images was proposed in [44]. Some NR-IQA methods using deep features have shown promising results on RGB images [25–31].

However, applying these NR-IQA methods to HSIs leads to many kinds of issues. Firstly, regarding the NR-IQA methods for pan-sharpening images, these methods require corresponding panchromatic images (PANs) and low-resolution multispectral images (LR-MSIs) as inputs, which are not available in HSI datasets. Secondly, regarding the NR-IQA methods for remote sensing images and RGB images, these methods require MOS labels for training neural networks and validating. Since humans cannot visually evaluate the quality of the spectrum, there is a lack of HSI datasets with MOS labels for HSI quality assessment.

### 2.1.3. Vision Transformer

Transformers were initially introduced for natural language processing (NLP) [45] and have achieved great success. Recently, the Transformer in computer vision has also demonstrated powerful modeling ability. Dosovitskiy et al. [46] proposed dividing an image into multiple windows, considering each window as a token, and computing Multi-Head Self-Attention (MSA) between these window tokens. Subsequently, Swin Transformer [47] introduced Windows-MSA, which partitions the feature map into multiple windows and computes Windows-MSA within each window.

However, these models are designed for RGB images and calculate self-attention scores along the spatial dimension. HSIs exhibit spectral similarity and correlation, making them more reasonable to use to calculate attention along the spectral dimension rather than the spatial dimension. The Mask-guided Spectral-wise Transformer (MST) [32], specifically designed for HSI reconstruction, utilizes S-MSA and has achieved impressive results. Inspired by S-MSA, we designed a S-Transformer to extract spectral-aware deep features that are more suitable for quality assessment rather than HSI reconstruction.

### 2.2. Method

#### 2.2.1. S-Transformer for Extracting Deep Features

NR-IQA methods for RGB images only focus on spatial features and neglect spectral features. We propose the S-Transformer to explore spectral information. Unlike MST [32], which adopts a U-shape architecture, S-Transformer reduces the size of feature maps and finally maps features to a rank score.

As shown in Figure 2a, given a 3D HSI cube $I \in \mathbb{R}^{H \times W \times C}$, $I$ firstly undergoes an embedding layer. Sequentially, in the first two stages, feature maps undergo a Spectral Attention Block (SAB) and a downsampling layer. The SAB layer does not change the shape of the input, while the downsampling layer reduces the spatial resolution by half and doubles the number of channels. In the last three stages, the downsampling layer only reduces the spatial resolution by half. After five stages, the shape of the feature map is $\frac{H}{32} \times \frac{W}{32} \times 4C$. The feature map will be flattened into a one-dimensional vector, which is then progressively mapped to $1 \times 1 \times 4096$, $1 \times 1 \times 4096$, $1 \times 1 \times 1000$, $1 \times 1 \times 512$, $1 \times 1 \times 64$, and $1 \times 1 \times 1$ vectors through fully connected layers and ReLU activation. The final output is considered the score for this HSIs.

As shown in Figure 2b, SAB first normalizes the batch data by layer normalization. For convenience of representation, the subsequent process is illustrated using a single image.

SAB splits the normalized input $X_{in} \in \mathbb{R}^{H \times W \times C}$ along the spectral dimension. Then, these vectors are flattened into $X \in \mathbb{R}^{HW \times C}$ and input into the S-MSA to calculate spectral self-attention.

$$Flatten(X_{in}(:,:,i)) = X(:,i) \tag{1}$$

Then, $X$ is linearly projected into value $V \in \mathbb{R}^{HW \times C}$, key $K \in \mathbb{R}^{HW \times C}$, and query $Q \in \mathbb{R}^{HW \times C}$:

$$V = XW^V, \ K = XW^K, \ Q = XW^Q \tag{2}$$

where $W^V$, $W^K$, and $W^Q \in \mathbb{R}^{C \times C}$ are learnable parameters. Subsequently, we split $V$, $K$, and $Q$ into $N$ heads along the spectral channel dimension, and the dimension of each head is $d_h = \frac{C}{N}$:

$$V = [V_1, \cdots, V_N], \ K = [K_1, \cdots, K_N], Q = [Q_1, \cdots, Q_N] \tag{3}$$

We calculate the attention score along spectral dimension:

$$A_j = \text{softmax}\left(\sigma_j K_j^T Q_j\right),\ head_j = V_j A_j \tag{4}$$

where $K_j^T$ represents the transpose matrix of $K_j$, $A_j$ represents self-attention and $\sigma_j \in \mathbb{R}^1$ represents a learnable parameter to adapt the weights of self-attention.

The outputs of $N$ heads are concatenated along a spectral dimension to undergo a linear projection. Finally, it is added with a position embedding $f_p(V)$:

$$S - MSA(X) = \underset{j=1}{\overset{N}{Concat}}\left(head_j\right)W + f_p(V) \tag{5}$$

where $W \in \mathbb{R}^{C \times C}$ represents the learnable parameters of the linear layer and $f_p(\cdot)$ is the function used to generate positional embedding.

The shape of the output of S-MSA is $H \times W \times C$. The outputs will undergo layer normalization again. After that, the outputs undergo three conv layers. The first one is a $1 \times 1$ Conv2d layer, and the shape of the output is $H \times W \times 4C$. The second one is a $3 \times 3$ Depth-Wise Conv2d layer, and the shape of the output is still $H \times W \times 4C$. The last one is a $1 \times 1$ Conv2d layer, and the shape of the output is $H \times W \times C$.

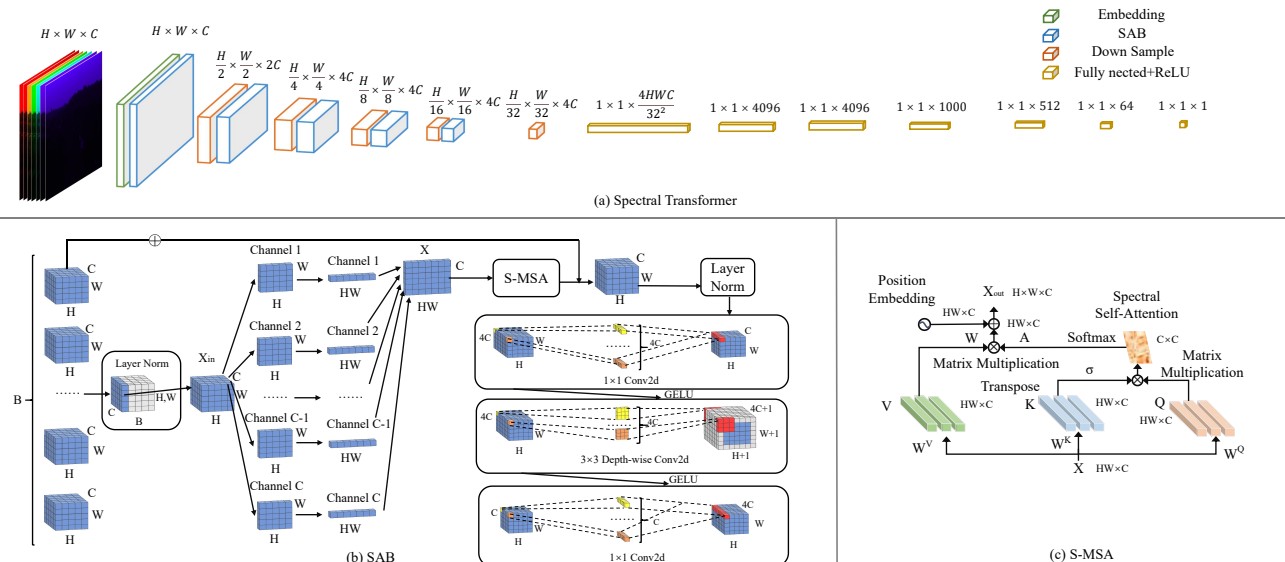

**Figure 2.** Network Architecture. (**a**) is the structure of S-Transformer. (**b**) is the structure of SAB. (**c**) is the structure of S-MSA.

### 2.2.2. Ranking Feature Learning for Pretraining

Ranking Feature Learning is designed to enable the network to accurately reflect the quality of two images that have different ranks of distortion. Gaussian noise and blurring are common distortion phenomena in HSIs [48–50]. Therefore, we consider HSIs with randomly added Gaussian noise with $\sigma = 0.05, 0.20$ as distortion images of noise ranks 1 and 2, and HSIs generated by $3 \times 3$, $5 \times 5$ blurring kernels as distortion images of blur ranks 1 and 2. The quality of images with lower ranks is better than those with higher ranks of the same distortion type.

As shown in Figure 3, we input images of different ranks in pairs into the network for scoring, aiming for the output scores of the network to be consistent with the real ranks. Therefore, we use pairwise ranking hinge loss to update the parameters of the network. Assuming $rank_{x_1} > rank_{x_2}$, the loss is given by:

$$L(x_1, x_2; \theta) = \max(0, f(x_1; \theta) - f(x_2; \theta) + \epsilon) \tag{6}$$

where $x_1$ and $x_2$ represent distortion HSIs of different distortion ranks, $\theta$ represents the parameters of the deep network, and $\epsilon$ is the margin used to ensure $f(x_1; \theta) \neq f(x_2; \theta)$. The gradient of the loss in (6) is given by:

$$\nabla_\theta L = \begin{cases} 0 & \text{if } f(x_2; \theta) - f(x_1; \theta) + \epsilon \geq 0, \\ \nabla_\theta f(x_1; \theta) - \nabla_\theta f(x_2; \theta) & \text{otherwise.} \end{cases} \tag{7}$$

When the outputs of the network match the real ranks, the gradient of the loss is zero. When the outputs of the network contradict real ranks, the gradient of the higher score is decreased while the gradient of the lower score is increased. However, this leads to a problem: each time the network parameters are updated, the network tends to output larger scores for higher-quality images. Through multiple epochs of training, although the loss value converges, the output values of the pre-trained network diverge, meaning that the magnitude of the output values keeps increasing. The network can ensure the right ranks of two images, but the output results cannot be directly used as a metric. The method we propose in the subsequent section can solve this problem.

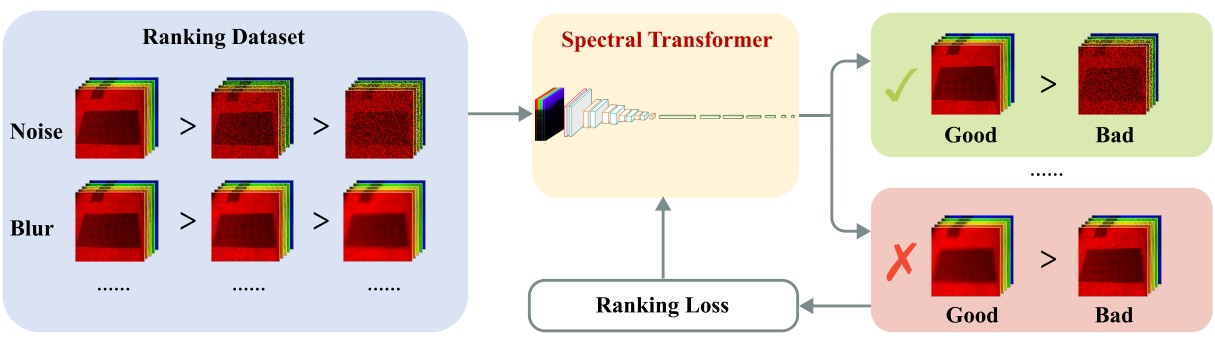

**Figure 3.** Process of pretraining task.

2.2.3. Wasserstein Distance for Measuring Non-Overlapping Distribution

We consider measuring the degree of image distortion using the discrepancy between the distribution of the deep features. We take the output $\mathcal{F} \in \mathbb{R}^{1 \times 1000}$ of the third linear layer in the pre-trained network as the deep features, which follows the basic form of a Gaussian distribution:

$$f(x) = \frac{1}{\sigma \sqrt{2\pi}} \exp\left[ -\frac{(x - \mu)^2}{2\sigma^2} \right] \tag{8}$$

Taking the average of the deep features of $M$ pristine HSIs yields $\overline{\mathcal{F}}$:

$$\overline{\mathcal{F}} = \sum_{i=1}^{M} \mathcal{F}_i \tag{9}$$

We consider the probability distribution $p(\overline{\mathcal{F}})$ of $\overline{\mathcal{F}}$ as the benchmark distribution.

For a reconstructed HSIs in the new scene without ground-truth HSIs, we can input it to the pre-trained network to obtain its deep feature $\mathcal{F}' \in \mathbb{R}^{1 \times 1000}$, with its probability distribution denoted as $q(\mathcal{F}')$.

We apply the Wasserstein distance to measure the discrepancy. The formula for calculating the Wasserstein distance between the reconstructed HSI deep feature distribution and the benchmark distribution is as follows:

$$\mathcal{W}\left[ p(\overline{\mathcal{F}}), q(\mathcal{F}') \right] = \inf_{\gamma \in \prod[p,q]} \iint \gamma(x, y) c(x, y) dx dy \tag{10}$$

where $c(x,y)$ represents the cost function for transforming $p$ into $q$, and $\gamma \in \prod[p,q]$ represents the joint distribution of $p$ and $q$:

$$c(x,y) = \|x - y\| \tag{11}$$

$$\int \gamma(x,y)dy = p(x) \tag{12}$$

$$\int \gamma(x,y)dx = q(y) \tag{13}$$

Compared to Kullback–Leibler (KL) divergence (14), the Jensen–Shannon (JS) divergence (15), and the Bhattacharyya distance (16), the Wasserstein distance is more suitable for measuring the discrepancy in non-overlapping cases.

$$D_{KL}(p||q) = \int p(x) \log \frac{p(x)}{q(x)} dx \tag{14}$$

$$D_{JS}(p||q) = D_{KL}\left( p \middle\| \frac{p+q}{2} \right) + D_{KL}\left( q \middle\| \frac{p+q}{2} \right) \tag{15}$$

$$D_B(p||q) = -\ln\left( \int \sqrt{p(x)q(x)}dx \right) \tag{16}$$

The following example illustrates the advantages of Wasserstein distance compared to other distances in non-overlapping situation. Consider two distributions $p_0$ and $p_\theta$ in the two-dimensional space as follows: as shown in Figure 4, $p_0$ is uniformly distributed along the unit line segment AB, and $p_\theta$ is uniformly distributed along the unit line segment CD. The distance between the two distributions can be controlled by the parameter $\theta$.

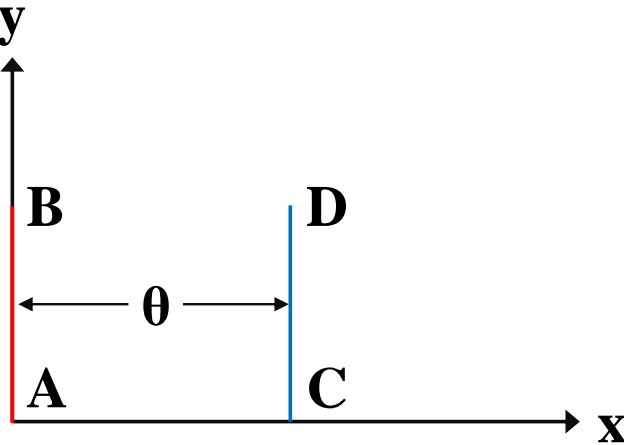

**Figure 4.** Two examples to illustrate the advantage of Wasserstein distance.

In the above scenario, the results of the aforementioned distances are as follows:

$$\mathcal{W}(p_0, p_\theta) = |\theta| \tag{17}$$

$$D_{KL}(p_0||p_\theta) = \begin{cases} 0, & \text{if } \theta = 0, \\ +\infty, & \text{if } \theta \neq 0. \end{cases} \tag{18}$$

$$D_{JS}(p_0||p_\theta) = \begin{cases} 0, & \text{if } \theta = 0, \\ \log 2, & \text{if } \theta \neq 0. \end{cases} \tag{19}$$

$$D_B(p_0||p_\theta) = \begin{cases} 0, & \text{if } \theta = 0, \\ +\infty, & \text{if } \theta \neq 0. \end{cases} \tag{20}$$

From the above equations, it is evident that Wasserstein distance is controlled by $\theta$ and retains meaningful values. In contrast, both KL divergence and Bhattacharyya distance yield $+\infty$ when $\theta \neq 0$, regardless of its variation, while JS divergence remains constant. Clearly, Wasserstein distance can better capture distributional differences.

## 3. Experiment Design and Results

### 3.1. Dataset and Experiment Setting

In HSI reconstruction, there is an assumption that the ground truth is a high-quality image, which is typically captured by high-quality imaging detectors under sufficient exposure and stable conditions. Therefore, in HSI reconstruction tasks, predictions that are closer to the ground truth are considered higher quality, indicative of better image fidelity. To demonstrate the effectiveness of the proposed assessment method, we test if the proposed quality scores are consistent with other FR-IQA metrics. We firstly apply seven state-of-the-art HSI reconstruction methods to the simulation dataset,s and then the quality scores of the reconstructed HSIs are computed and compared with FR-IQA metrics to see if there is consistency.

The ARAD_1K dataset [51] provides a total of 950 HSIs named as ARAD_0001 to ARAD_0950. These HSIs have a spatial dimension of 482 × 512 and consist of 31 spectral bands ranging from 400 nm to 700 nm. For the pretraining task, we utilize a subset of 900 HSIs from ARAD_0001 to ARAD_0900 as the pristine HSI dataset. The Gaussian noise ranking dataset is created by adding random Gaussian noise with $\sigma = 0.05, 0.20$ to the pristine HSIs. The blurring ranking dataset is formed by applying 3 × 3 and 5 × 5 blurring kernels. Differently ranked HSIs of the same distortion type (including the pristine image) are paired and input into the pretraining network, such as ARAD_0001 and noise_1_ARAD_0001, and blur_1_ARAD_0001 and blur_2_ARAD_0001. These HSI pairs are divided into training and validation datasets in an 8:1 ratio.

Subsequently, to verify the consistency between R-NHSIQA and FR-IQA metrics and compare them with the QSFL [24], we conducted simulation experiments on 50 scenes from ARAD_0901 to ARAD_0950. We compared the assessment scores of the reconstructed HSIs obtained from seven representative SOTA HSI reconstruction algorithms [7–12,32] as follows:

- lambda-Net (denoted as $\lambda$-Net) [7].
- Deep tensor admm-net (denoted as ADMM-Net) [8].
- High-resolution dual-domain learning for spectral compressive imaging (denoted as HDNet) [9].
- Mask-guided Spectral-wise Transformer (denoted as MST) [32].
- Coarse-to-fine sparse transformer (denoted as CST) [10].
- Degradation-Aware Unfolding Half-Shuffle Transformer (denoted as DAUHST) [11].
- Residual Degradation Learning Unfolding Framework-MixS$^2$ Transformer (denoted as RDLUF-MixS$^2$) [12].

All the experiments are implemented on Pycharm Community 2022, with NVIDIA GeForce RTX 3090, RAM of 32 GB. The training of our method takes about 4 h, while assessing 350 reconstructed HSIs takes about 60 s.

### 3.2. Evidence of Quality Sensitivity of Our Deep Features

We visualized the distribution of the mean and the standard deviation of the deep features from 900 pristine and distorted HSIs. The results in Figure 5 indicate that different degrees of distortion significantly shift the distribution of the mean and standard deviation of HSI deep features from the prestine HSIs.

We calculated the average of the deep features for 900 HSIs of the same distortion type and plotted the Gaussian distribution curve of the average deep features as shown in Figure 6. From Figure 6, it can be observed that the overlap area between distortion distribution and pristine distribution decreases as the rank of distortion increases.

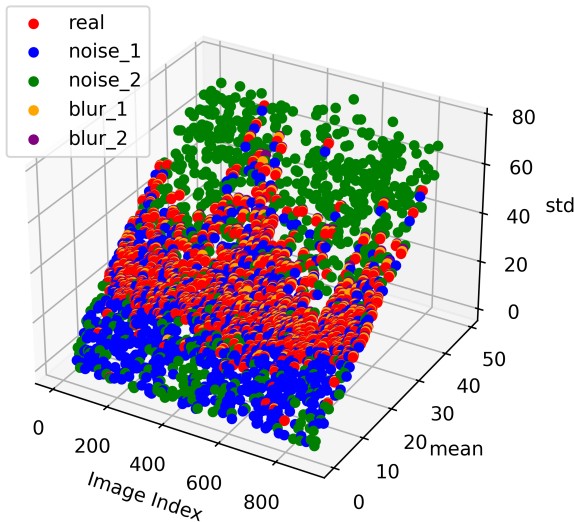

**Figure 5.** The distribution parameters of the deep features for 900 HSIs of different types of distortion.

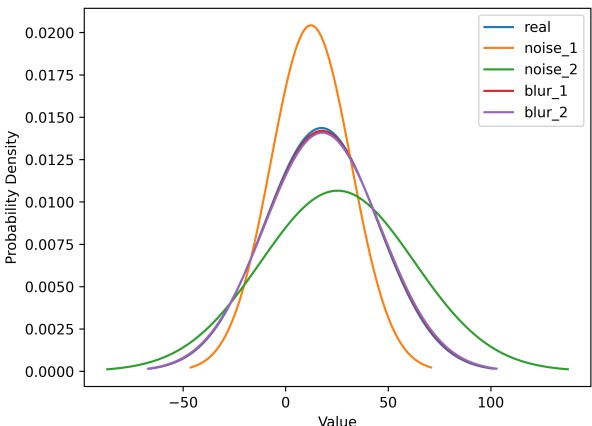

**Figure 6.** The Gaussian distribution curves of average deep features for different types of distortion.

We evaluated the distorted HSIs of different ranks using Wasserstein distance, and the results are shown in Table 1. From Table 1, the Wasserstein distance between the distrubution of the deep features and the benchmark distribution increases as the rank of distortion increases. This result validates the effectiveness of using Wasserstein distance as the evaluation score.

**Table 1.** The Wasserstein distance between different types of distortion distributions and the benchmark distribution.

| Noise_1 | Noise_2 | Blur_1 | Blur_2 |
|---|---|---|---|
| 5.2114639 | 8.3326150 | 0.2056402 | 0.3490431 |

### *3.3. Consistency between R-NHSIQA, QSFL, and FR-IQA Metrics*

In Table 2, the average scores of R-NHSIQA, QSFL [24], and FR-IQA metrics [34,35,37,38] for reconstructed HSIs of different reconstruction algorithms are reported. From Table 2, it can be observed that R-NHSIQA is consistent with all FR-IQA metrics, while QSFL fails with some reconstruction algorithms. Specifically, QSFL scored the second highest on the

reconstruction results of ADMM-Net, while the FR-IQA metrics of ADMM-Net were lower than those of most reconstruction algorithms. To provide a visual comparison, we have also plotted Figures 7 and 8 to demonstrate the consistency between R-NHSIQA and FR-IQA metrics, as well as between QSFL and FR-IQA metrics.

**Table 2.** The average scores of the reconstructed HSIs from different reconstruction algorithms.

| Metrics | ADMM-Net | CST | DAUHST | $\lambda$-Net | RDLUF-MixS$^2$ | HDNet | MST |
|---|---|---|---|---|---|---|---|
| PSNR ↑ | 32.42039 | 34.90362 | 36.87901 | 28.18398 | 38.08767 | 33.94883 | 34.05283 |
| SSIM [34]↑ | 0.85144 | 0.92686 | 0.94296 | 0.74009 | 0.95781 | 0.91139 | 0.91672 |
| Q2$^n$ [38]↑ | 0.63312 | 0.71071 | 0.77288 | 0.44178 | 0.81445 | 0.66966 | 0.69030 |
| MvSSIM [35]↑ | 0.89628 | 0.95191 | 0.96830 | 0.73761 | 0.97485 | 0.93707 | 0.93725 |
| SAM [37]↓ | 14.73825 | 9.15665 | 6.82514 | 31.08199 | 4.83432 | 11.67378 | 10.41821 |
| QSFL [24]↓ | 22.43255 | 22.63995 | 22.71708 | 37.24140 | 22.58776 | 23.25978 | 21.93860 |
| R-NHSIQA↓ | 3.18004 | 2.25054 | 1.77165 | 7.55191 | 1.50417 | 2.79436 | 2.63320 |

The uparrow indicates that a higher value is better, while the downarrow signifies that a smaller value is better. The same principle applies to the subsequent table.

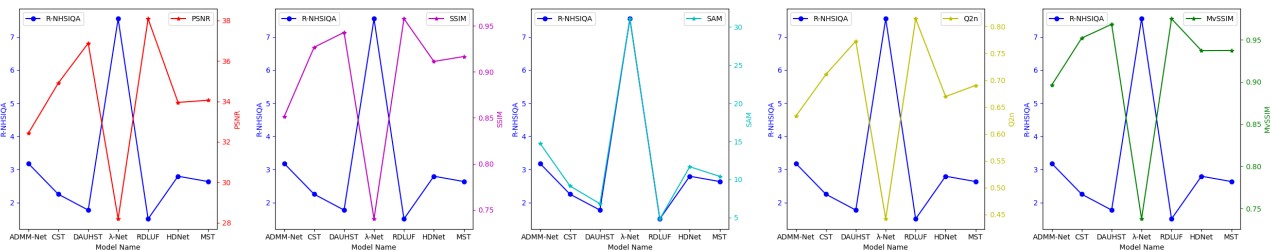

**Figure 7.** Consistency between R-NHSIQA and FR-IQA in the evaluation of different reconstruction algorithms based on the average scores.

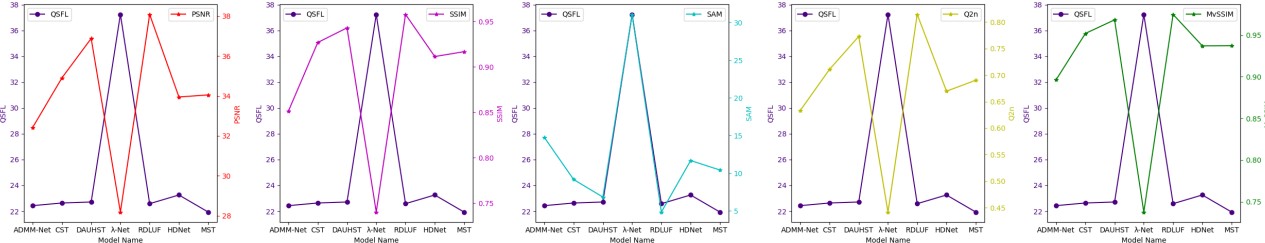

**Figure 8.** Consistency between QSFL and FR-IQA in the evaluation of different reconstruction algorithms based on the average scores.

In addition, we have also reported the Spearman rank order correlation coefficient (SROCC), Kendall rank order correlation coefficient (KROCC), and Pearson correlation coefficient (PLCC) of the proposed method and QSFL with different FR-IQA metrics. Equations (21)–(23) present the formulas for calculating correlation coefficients.

$$SROCC = 1 - \frac{6\sum_{i=1}^{n} d_i^2}{n(n^2-1)} \tag{21}$$

where $n$ is the number of samples, $d_i$ is the rank difference between the R-NHSIQA score and FR-IQA score of the $i$-th sample.

$$KROCC = \frac{2(C-D)}{n(n-1)} \tag{22}$$

where $C$ is the number of concordant pairs and $D$ is the number of discordant pairs.

$$PLCC = \frac{\sum_{i=1}^{n}(x_i - \overline{x})(y_i - \overline{y})}{\sqrt{\sum_{i=1}^{n}(x_i - \overline{x})^2 \sum_{i=1}^{n}(y_i - \overline{y})^2}} \qquad (23)$$

where $x_i$ is the R-NHSIQA score of the $i$-th sample and $y_i$ is the FR-IQA score of the $i$-th sample.

The range of values for SROCC, KROCC, and PLCC is $[-1, 1]$, with a value of 1 indicating positive correlation, a value of $-1$ indicating negative correlation, and a value of 0 indicating uncorrelated pairs of variables. As shown in Tables 3–7, the proposed R-NHSIQA achieves stronger consistency with various FR-IQA metrics on all correlation coefficients compared to QSFL.

**Table 3.** The correlation coefficients between PSNR and R-NHSIQA on reconstructed HSIs.

| Metrics | SROCC↓ | KROCC↓ | PLCC↓ |
|---|---|---|---|
| QSFL [24] | −0.43157 | −0.27368 | −0.49587 |
| R-NHSIQA | −0.79098 | −0.63158 | −0.64128 |

**Table 4.** The correlation coefficients between SSIM and R-NHSIQA on reconstructed HSIs.

| Metrics | SROCC↓ | KROCC↓ | PLCC↓ |
|---|---|---|---|
| QSFL [24] | −0.56090 | −0.31578 | −0.48856 |
| R-NHSIQA | −0.75188 | −0.54737 | −0.69440 |

**Table 5.** The correlation coefficients between SAM and R-NHSIQA on reconstructed HSIs.

| Metrics | SROCC↑ | KROCC↑ | PLCC↑ |
|---|---|---|---|
| QSFL [24] | 0.60150 | 0.44210 | 0.48292 |
| R-NHSIQA | 0.77669 | 0.60526 | 0.71525 |

**Table 6.** The correlation coefficients between $Q2^n$ and R-NHSIQA on reconstructed HSIs.

| Metrics | SROCC↓ | KROCC↓ | PLCC↓ |
|---|---|---|---|
| QSFL [24] | −0.61127 | −0.48421 | −0.52506 |
| R-NHSIQA | −0.74098 | −0.53421 | −0.74859 |

**Table 7.** The correlation coefficients between MvSSIM and R-NHSIQA on reconstructed HSIs.

| Metrics | SROCC↓ | KROCC↓ | PLCC↓ |
|---|---|---|---|
| QSFL [24] | −0.54135 | −0.30526 | −0.33847 |
| R-NHSIQA | −0.76616 | −0.59473 | −0.70100 |

Figures 9 and 10 display the pseudo-RGB images of the reconstructed HSIs and evaluation scores of different FR-IQA and NR-IQA metrics of two scenes. In Figures 9 and 10, the proposed R-NHSIQA is not only consistent with all the FR-IQA metrics but also consistent with visual perception. In contrast, QSFL is inconsistent with FR-IQA metrics except for $\lambda$-Net. Specifically, in Figure 9, although the quality of the HSIs reconstructed by MST is significantly lower than that of RDLUF-MixS$^2$, MST still achieves the best QSFL score.

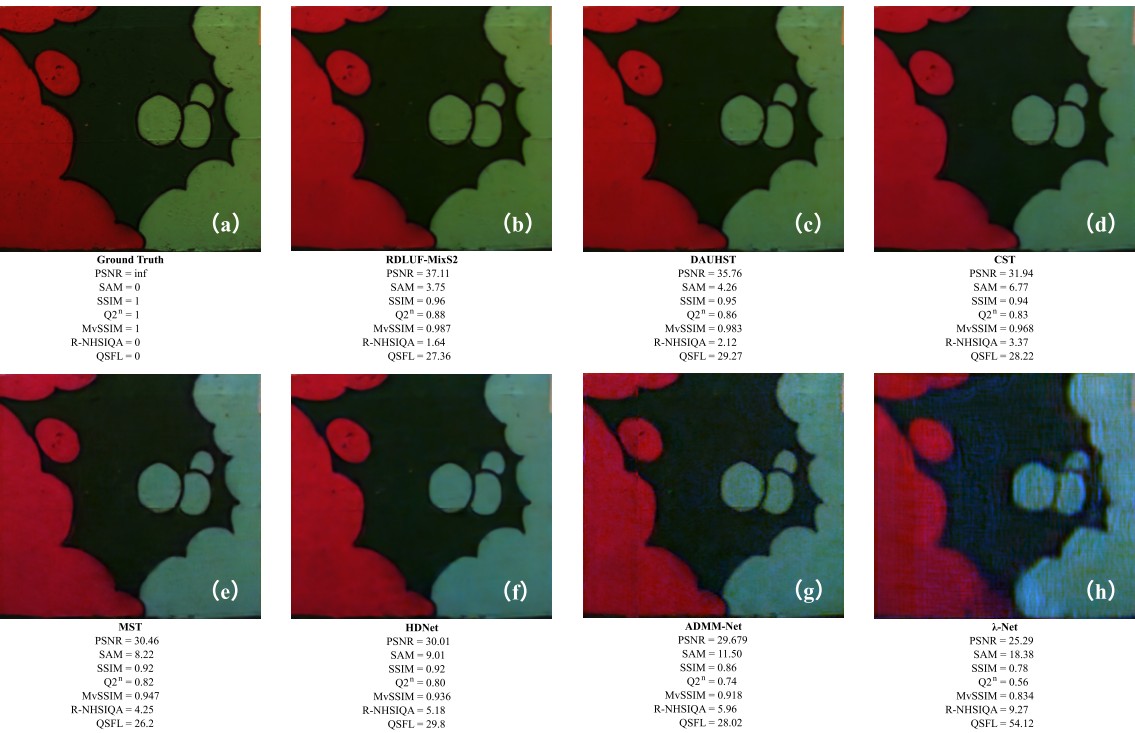

**Figure 9.** The assessment scores of a sample scene. The pristine HSIs and reconstructed HSIs are shown in RGB. (**a**) is the pseudo-RGB image of the pristine HSI, and (**b**–**h**) are respectively the pseudo-RGB images of the HSIs reconstructed by RDLUF-MixS$^2$, DAUHST, CST, MST, HDNet, ADMM-Net, $\lambda$-Net. The quality scores of these images are shown below.

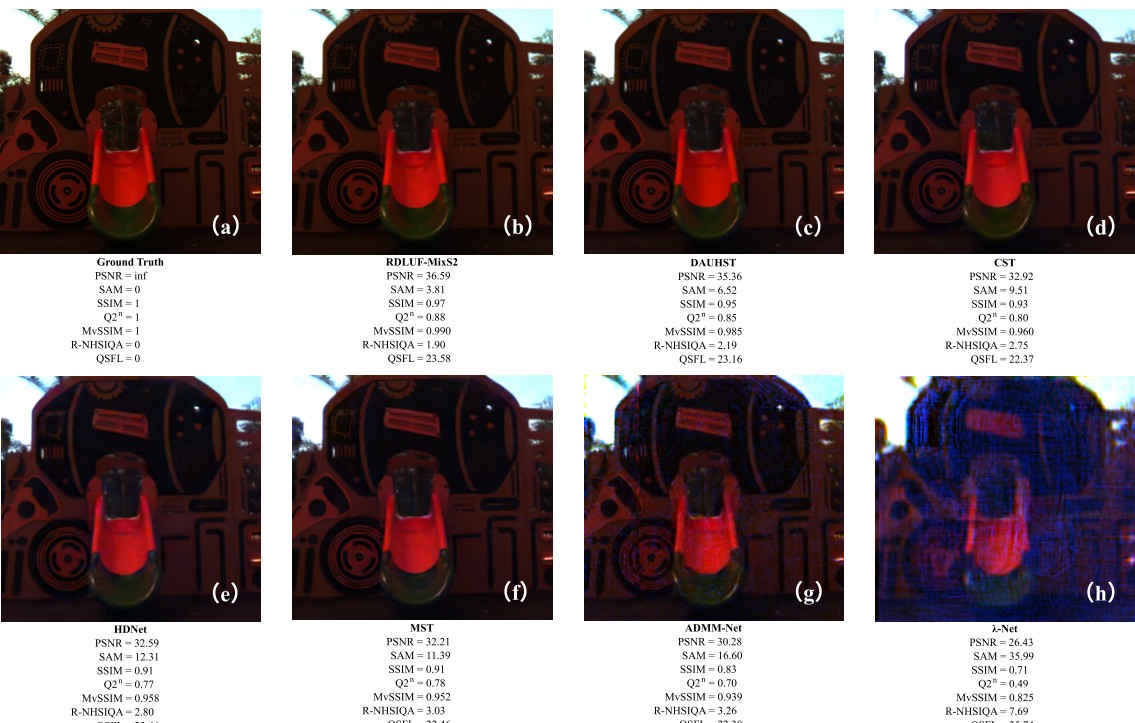

**Figure 10.** The assessment scores of a sample scene. The pristine HSIs and reconstructed HSIs are shown in RGB. (**a**) is the pseudo-RGB image of the pristine HSI, and (**b**–**h**) are respectively the pseudo-RGB images of the HSIs reconstructed by RDLUF-MixS$^2$, DAUHST, CST, MST, HDNet, ADMM-Net, $\lambda$-Net. The quality scores of these images are shown below.

### 3.4. Comparison with Different Feature Extracting Networks

To demonstrate the suitability of the S-Transformer for extracting features from HSIs, we compared its performance with other network structures in this section. VGG16 [52] is a classic CNN network and is employed as a replacement for the S-Transformer in the feature extraction. The deep features in the fc7 layer are used to calculate the Wasserstein distance.

The VGG16 has been pre-trained via ranking feature learning. Figure 11 and Table 8 show that the deep features extracted by VGG16 also exhibit sensitivity to quality. However, compared to Figure 6, the differences between different distortions are less obvious, which indicates that the proposed S-Transformer can utilize spectral information to produce more representative deep features.

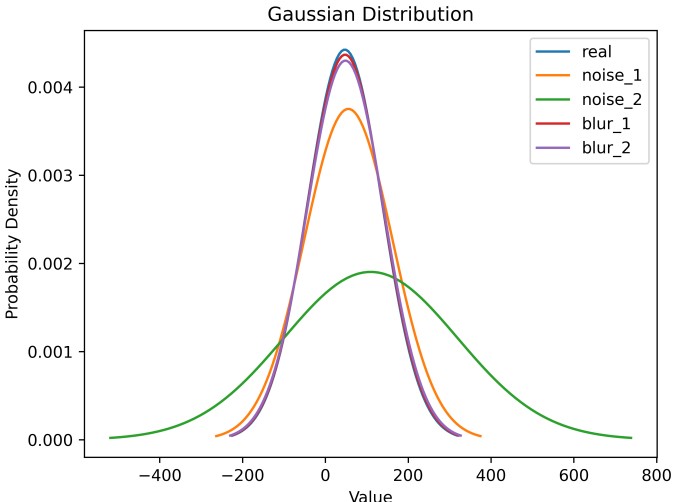

**Figure 11.** The Gaussian distribution curves of average deep features extracted by VGG16 for different types.

**Table 8.** The Wasserstein distance between different types of distortion distributions and the benchmark distribution when the network is VGG16.

| noise_1 | noise_2 | blur_1 | blur_2 |
|---------|---------|--------|--------|
| 8.4404264 | 62.4436738 | 0.6140266 | 1.3637162 |

In Tables 9–14 and Figure 12, we have shown the scores of R-NHSIQA with different feature extracting networks and FR-IQA metrics. Compared to Tables 2–7 and Figure 7, the consistency between R-NHSIQA with VGG16 and FR-IQA metrics decreased. This could be attributed to traditional CNNs failing to fully utilize spectral information, resulting in the loss of such information in the extracted deep features.

**Table 9.** Comparison of R-NHSIQA using different networks.

| Metrics | ADMM-Net | CST | DAUHST | $\lambda$-Net | RDLUF-MixS$^2$ | HDNet | MST |
|---------|----------|-----|--------|--------|-----------|-------|-----|
| **PSNR↑** | 32.42039 | 34.90362 | 36.87901 | 28.18398 | 38.08767 | 33.94883 | 34.05283 |
| **SSIM [34]↑** | 0.85144 | 0.92686 | 0.94296 | 0.74009 | 0.95781 | 0.91139 | 0.91672 |
| **Q2$^n$ [38]↑** | 0.63312 | 0.71071 | 0.77288 | 0.44178 | 0.81445 | 0.66966 | 0.69030 |
| **MvSSIM [35]↑** | 0.89628 | 0.95191 | 0.96830 | 0.73761 | 0.97485 | 0.93707 | 0.93725 |
| **SAM [37]↓** | 14.73825 | 9.15665 | 6.82514 | 31.08199 | 4.83432 | 11.67378 | 10.41821 |
| **Ours (VGG16)↓** | 2.51364 | 2.56315 | 2.41836 | 3.35436 | 2.33156 | 2.64563 | 2.47669 |
| **Ours (S-Transformer)↓** | 3.18004 | 2.25054 | 1.77165 | 7.55191 | 1.50417 | 2.79436 | 2.63320 |

**Table 10.** The correlation coefficients between PSNR and R-NHSIQA using different networks on reconstructed HSIs.

| Metrics | SROCC↓ | KROCC↓ | PLCC↓ |
|---|---|---|---|
| **w/VGG16** | −0.64552 | −0.48631 | −0.50153 |
| **w/S-Transformer** | −0.79098 | −0.63158 | −0.64128 |

**Table 11.** The correlation coefficients between SSIM and R-NHSIQA using different networks on reconstructed HSIs.

| Metrics | SROCC↓ | KROCC↓ | PLCC↓ |
|---|---|---|---|
| **w/VGG16** | −0.62685 | −0.45210 | −0.48303 |
| **w/S-Transformer** | −0.75188 | −0.54737 | −0.69440 |

**Table 12.** The correlation coefficients between SAM and R-NHSIQA using different networks on reconstructed HSIs.

| Metrics | SROCC↑ | KROCC↑ | PLCC↑ |
|---|---|---|---|
| **w/VGG16** | 0.63541 | 0.46896 | 0.56314 |
| **w/S-Transformer** | 0.77669 | 0.60526 | 0.71525 |

**Table 13.** The correlation coefficients between $Q2^n$ and R-NHSIQA using different networks on reconstructed HSIs.

| Metrics | SROCC↓ | KROCC↓ | PLCC↓ |
|---|---|---|---|
| **w/VGG16** | −0.62563 | −0.49511 | −0.54630 |
| **w/S-Transformer** | −0.74098 | −0.53421 | −0.74859 |

**Table 14.** The correlation coefficients between MvSSIM and R-NHSIQA using different networks on reconstructed HSIs.

| Metrics | SROCC↓ | KROCC↓ | PLCC↓ |
|---|---|---|---|
| **w/VGG16** | −0.61335 | −0.47895 | −0.52604 |
| **w/S-Transformer** | −0.76616 | −0.59473 | −0.70100 |

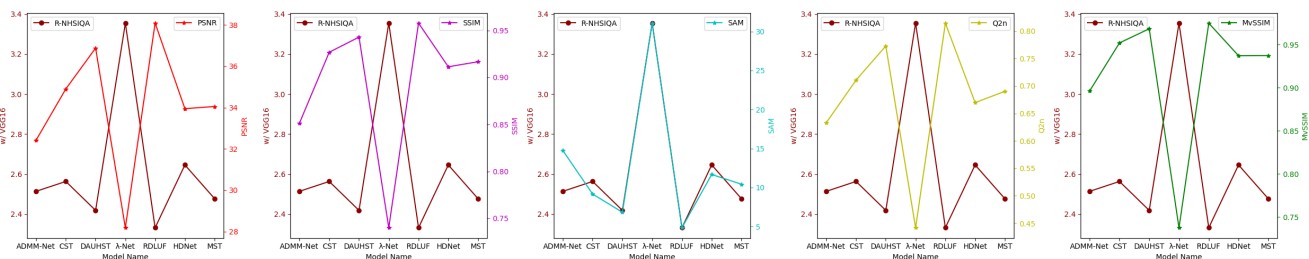

**Figure 12.** Consistency between R-NHSIQA using VGG16 and FR-IQA in the evaluation of different reconstruction algorithms based on the average scores.

## 4. Discussion

In this study, we propose a new non-reference HSI quality assessment metric. From the experimental results, we can make the following discussions:

- From the results in Sections 3.2 and 3.4, it can be observed that the deep neural network trained with the ranking feature learning task could capture feature distributions highly related to the degree of image distortion. Therefore, the extent of the deviation in feature distribution from the benchmark distribution could indicate the quality of reconstructed image.

- Extracting features from both spectral and spatial domains is more indicative of the quality of HSIs compared to extracting features solely from the spatial domain. For instance, as shown in Section 3.3, when using the S-Transformer, which simultaneously extracts spatial and spectral information, the quality score of the reconstructed image is consistent with objective metrics such as SAM. However, when using VGG16, which only operates in the spatial dimension, the quality score of the reconstructed image is not entirely consistent with FR-IQA metrics such as SAM.
- The proposed method evaluates image quality faster and is more consistent with FR-IQA metrics compared to previous manual-feature-based methods, such as QSFL [24]. QSFL required about 10 min to evaluate 350 reconstructed HSIs, while our method takes about 60 s. Furthermore, our method eliminates the cumbersome process of manual features. Lastly, as shown by the results in Section 3.3, our method achieves a stronger correlation with FR-IQA metrics compared to QSFL.

## 5. Conclusion and Limitation

In real hyperspectral imaging processes, due to the lack of ground truth, the best model validated on the simulation dataset via FR-IQA metrics may fail in real imaging processes. Therefore, it is crucial to design a suitable no-reference HSI quality assessment metric as a substitute for FR-IQA metrics to reflect the reconstruction performance of different models. To address this challenge, we propose a no-reference HSI quality assessment method via ranking feature learning (R-NHSIQA). Firstly, we proposed the S-Transformer to extract representative deep features, which leverages Spectral-wise Multi-head Self-Attention (S-MSA) to capture the significant spectrum information in HSIs. To extract quality-sensitive deep features, we employ the ranking feature learning as the pre-training task for the S-Transformer. Through ranking feature learning, the S Transformer is able to discern image quality. Additionally, we utilize the Wasserstein distance between the distribution of the deep features of the reconstructed HSIs and the benchmark distribution as the quality score, which could better measure the discrepancy between distributions even in non-overlapping situations. The experimental results indicate that the proposed R-NHSIQA shows good consistency with the FR-IQA metrics, validating the idea that the proposed metric can serve as a substitute for FR-IQA metrics in real-world tasks.

**Limitations:** Although the proposed method has made progress in no-reference HSI quality assessment, there are still some issues that could be improved upon in future work. For example, during rank feature learning, we have considered Gaussian noises and blurs. However, the real world has all kinds of unmodeled noises; thus, the proposed method might fail to these noises. As we deepen the understanding of the physical models of noises, the proposed method could incorporate these noises into the ranking pre-training in the future.

**Author Contributions:** Methodology, validation, formal analysis, writing—original draft, and visualization: Y.L.; conceptualization, methodology, investigation, writing—review and editing, and supervision: Y.D.; resources, supervision, project administration, and writing—review and editing: H.L.; resources, supervision, and project administration: D.L.; resources, supervision, and project administration: F.X.; and funding acquisition, resources, supervision, and project administration: D.G. All authors have read and agreed to the published version of the manuscript.

**Funding:** This research was funded by the National Key Research and Development Program of China OF FUNDER under grant number 2019YFA0706604, the Natural Science Foundation (NSF) of China under grant number 62205260, the Natural Science Foundation (NSF) of China under grant number 62293483, the Guangzhou Key Laboratory of Scene Understanding and Intelligent Interaction under grant number 202201000001, the project of Pazhou Lab (Huangpu) under grant number 2022K0904, the Fundamental Research Funds for the Central Universities under grant number YJSJ24006, China Scholarship Council under Grant Number 202306960040.

**Data Availability Statement:** Data associated with this research are available online. The ARAD_1K dataset is available at https://github.com/boazarad/ARAD_1K (accessed on 6 May 2024).

**Conflicts of Interest:** The authors declare no conflicts of interest.

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
