# Peer review of "No-Reference Hyperspectral Image Quality Assessment via Ranking Feature Learning"

_remotesensing, doi:10.3390/rs16101657_

Round 1
Reviewer 1 Report (Previous Reviewer 2)
Comments and Suggestions for Authors
The provided answers to the reviewer's report are comprehensive. The paper can be accepted.
Comments on the Quality of English LanguageAcceptable
Author Response
Dear reviewer,
Thank you so much for your reviewing! We deeply appreciate your recognition of our research work.
Reviewer 2 Report (Previous Reviewer 3)
Comments and Suggestions for Authors
The authors have improved the technical quality of the work and clarified most of my doubts.However, a few doubts remain. Therefore, I recommend minor revision. My comments are-
1. The work quantifies the image quality by comparing the original image with the reconstructued image. However, general IQA assessment approaches compare do not use this concept. So what is the rationale behind the proposed work?
Author Response
Please find the attachment.
This manuscript is a resubmission of an earlier submission. The following is a list of the peer review reports and author responses from that submission.
Round 1
Reviewer 1 Report
Comments and Suggestions for Authors
This paper introduces a No-reference Hyperspectral Imaging (HSI) Quality Assessment metric, presenting a well-organized structure and experiments that validate the efficacy of the proposed method. However, several crucial considerations need attention.
1.Are there any visual examples to show how the score relates to the image?
2.Is there any comparison with other non-reference image calculation methods?
3.What is the difference between the Spectral Transformer and other Transformers?
4.For making the experiment richer, it is better to use more noise for comparison experiments.
5.The experiment chose to pick images randomly, but the reviewer considered that it may be too few to choose only 10 images?
Comments on the Quality of English LanguageThis paper introduces a No-reference Hyperspectral Imaging (HSI) Quality Assessment metric, presenting a well-organized structure and experiments that validate the efficacy of the proposed method. However, several crucial considerations need attention.
1.Are there any visual examples to show how the score relates to the image?
2.Is there any comparison with other non-reference image calculation methods?
3.What is the difference between the Spectral Transformer and other Transformers?
4.For making the experiment richer, it is better to use more noise for comparison experiments.
5.The experiment chose to pick images randomly, but the reviewer considered that it may be too few to choose only 10 images?
Reviewer 2 Report
Comments and Suggestions for Authors
In the paper a no-reference HSI Quality Assessment (IQA) is proposed: Comments:
1. The literature review lacks methods designed for no reference image quality assessment of remote sensing images or pan-sharpened images, see Google Scholar: https://scholar.google.com/scholar? as_sdt=0%2C5&q=no+reference+image+quality+assessment+of+remote+sensing+images&btnG=
https://scholar.google.com/scholar? as_sdt=0%2C5&q=no+reference+image+quality+assessment+of+Pan-Sharpened+Images&btnG=
2. The last contribution with the Wasserstein distance should be rewritten as the justification is not a part of the contributions list.
3. More methods for no reference IQA that use ranking, like [16] should be included in related works.
4. The abbreviation of the Hyperspectral images (HSI) is defined in the paper several times. Please leave only those in the abstract and first sentence of Section I.
5. The used dataset does not contain scores obtained in tests with human subjects. This poses a problem since perceptual methods (developed for databases that contain such scores) will be compared with the approach whose capability to capture the subjectivity of human perception cannot be reliably determined. A solid discussion is needed on whether the used methodology and the dataset are valid for creating no reference method. Please justify the “measuring the degree of image distortion using the discrepancy between the distribution of deep features.” Is this a valid way to determine the reliable perceptual quality of images?
6. Figure 3 should be removed from the paper as it represents common knowledge.
7. The approach is not compared with other no-reference approaches, either designed for similar data or natural images, as described in the introduction. There is also a lack of deep learning approaches or simply fine-tuned transformer examples. Furthermore, full reference SSIM and PSNR cannot be seen as state-of-the-art approaches. Hence, the contribution of this study cannot be determined.
8. To evaluate IQA methods, correlation coefficients between objective scores and MOS are used. Hence, the mean values in Table 2 are not meaningful. Please refer to relevant literature. The connected points in images 7-8 also seem wrong. What do they indicate? It is written that “It can be observed that as the PSNR and SSIM increase, our score decreases, and as the SAM increases, our score also increases.” This means that authors try to replace widely used correlation coefficients (SRCC, PLCC, KRCC) or error (RMSE). Please use the correct evaluation methodology.
9. Captions of Fig. 7-8 should be rewritten, please look at Fig. 9.
10. Figures 9 and 10 should have obtained objective scores written below. Now, their presence requires a more thorough examination by a reader. Why are they displayed and commented on using only one sentence? Are they representative taking into account the size of the dataset (950 images)?
11. The readers cannot replicate the results. One of the links to Google Drive does not work. Also, personal drives should not be used for storing codes or datasets of materials shared along with a paper due to their fragility. Use a public repository instead.
12. The limitations of the study and the lessons learned should be included in the conclusions.
Acceptable language quality.
Reviewer 3 Report
Comments and Suggestions for Authors
1. The work mentioned that the unavailability of ground truth HIS is a problem for HIS quality assessment. However, for proper quality assessment you need HSI data with label about the quality of the images. Therefore, please rewrite the corresponding part.
2. Please discuss how RS image quality assessment is different than traditional IQA.
3. The literature review seems a bit weak. Please discuss about recent works on remote sensing/ hyperspectral image quality assessment.
a. Feature extraction approach for quality assessment of remotely sensed hyperspectral images
b.
4. What is SAB layer? Why is it used? How are the parameters of the SAB trained? There is no loss function in the network shown in Fig. 1. A, and b. In that case, how the weights of the conv layers trained?
5. It is unclear how the whole network works. I suggest you to include a complete schematic diagram of the network.
6. What is MOS labels mentioned in Page 6?
7. Why did the authors consider the output of the third linear layer in the pre-trained network as the deep features follow the basic form of a Gaussian distribution. I don’t find any reason why it should be Gaussian.
8. The authors mentioned- “The following are the HSI reconstruction algorithms we employed for the simulation. These methods were chosen because they can represent the SOTA performance in the 227
reconstruction field at different stages.”
However, HSI reconstruction is related to image enhancement/denoising. These methods do not explicitly quantify HSI data quality. Therefore, I am confused About why these methods were used to compare. I think traditional no-referenced IQA measures such as FSIM, SSIM, BRISQUE scores will be more suitable. Please check the paper on feature extraction approach for HSI quality assessment.
Comments on the Quality of English LanguageQuality of English can be slightly improved
Round 2
Reviewer 2 Report
Comments and Suggestions for Authors
The answers to the reviewer’s comments lack quality, do not address several questions, are chaotically arranged, cover several issues with only one sentence, and omit crucial parts. Also, evidence that the method is compared with any other learning-based approach should be explicitly shown in the paper.
Furthermore, some answers lead to other questions. For example, the authors wrote “Therefore, our metric can serve as a substitute for FR-IQA metrics in real-world tasks without ground truth.”. Hence, the introduction of NR-IQA measures requires justification. Why the FR measures are not introduced in this paper in a devoted subsection?
Requires proofreading, e.g., lack comma in "5% to 15%, 25% to 35% and 45% to 55%"
Reviewer 3 Report
Comments and Suggestions for Authors
The authors addressed most of my queries. However, some doubts and suggestions still remain. Therefore, I recommend a major revision. The main observations are listed below-
-
Why does the work use Wasserstein distance for quality assessment?
-
The work mentioned- “However, applying them directly to HSI would encounter two issues. Firstly, these methods can only evaluate spatial quality and cannot measure spectral fidelity, which is crucial for HSI. Secondly, the training of these methods typically relies on RGB datasets with the Mean Opinion Score (MOS) as the predicted label”.
However, no reference QA does not require any training. Only referenced QA requires training. Please explain.
-
The work did not discuss about the existing works on hyperspectral image quality assessment. It needs to be improved. Please discuss the following works-
Feature extraction approach for quality assessment of remotely sensed hyperspectral images
No-reference hyperspectral image quality assessment via quality-sensitive features learning
Hypercomplex quality assessment of multi/hyperspectral images
MvSSIM: A quality assessment index for hyperspectral images
Comments on the Quality of English LanguageThe language can be slightly improved make the paper more lucid
